# A Systematic Review of Reported Cases of Immune Thrombocytopenia after COVID-19 Vaccination

**DOI:** 10.3390/vaccines10091444

**Published:** 2022-09-01

**Authors:** Prachi Saluja, FNU Amisha, Nitesh Gautam, Harmeen Goraya

**Affiliations:** 1Department of Internal Medicine, University of Arkansas for Medical Sciences, Little Rock, AR 72211, USA; 2Division of Pulmonary and Critical Care Medicine, Department of Internal Medicine, University of Arkansas for Medical Sciences, Little Rock, AR 72211, USA

**Keywords:** COVID-19 vaccine, BNT162b2 vaccine, mRNA-1273 vaccine, Ad26.COV2-S vaccine, ChAdOx1 nCoV-19 vaccine, thrombocytopenia, immune thrombocytopenia, ITP, immune thrombocytopenic purpura, idiopathic thrombocytopenic purpura

## Abstract

With the recent outbreak of the COVID-19 pandemic and emergency use authorization of anti-SARS-CoV-2 vaccines, reports of post-vaccine immune thrombocytopenia (ITP) have gained attention. With this systematic review, we aim to analyze the clinical characteristics, therapeutic strategies, and outcomes of patients presenting with ITP after receiving COVID-19 vaccination. Medline, Embase, and Ebsco databases were systematically explored from inception until 1 June 2022. Case reports and case series investigating the association between the anti-SARS-CoV-2 vaccine and ITP were included. We found a total of 66 patients. The mean age of presentation was 63 years with a female preponderance (60.6%). Sixteen patients had pre-existing ITP. The mean time from vaccine administration to symptom onset was 8.4 days. More ITP events were triggered by mRNA vaccines (BNT162b2 (*n* = 29) > mRNA-1273 (*n* = 13)) than with adenoviral vaccines (ChAdOx1-S AstraZeneca (*n* = 15) > Ad26.COV2-S (*n* = 9)). Most of the patients were treated with steroids or IVIG, or both. The overall outcome was promising, with no reported deaths. Our review attempts to increase awareness among physicians while evaluating patients presenting with thrombocytopenia after receiving the vaccine. In our solicited opinion, the rarity of these events and excellent outcomes for patients should not change views regarding the benefits provided by immunization.

## 1. Introduction

Immune thrombocytopenia (ITP) is an autoimmune disorder characterized by low platelet counts (<100 × 10^3^/μL) unexplainable by an alternative etiology [1]. ITP carries an annual incidence of about 3 cases per 100,000 adults, with a predilection for the female gender in the younger population [2]. Although patients may be asymptomatic upon presentation, typical clinical features include mucocutaneous bleeding, such as petechiae; purpura; ecchymoses; and sometimes, hemorrhage, with intracranial being the most serious [1,3]. While primary ITP is idiopathic in origin, secondary ITP can be caused by other autoimmune disorders, cancer, infection, or medications and accounts for less than a fourth of total ITP cases [4]. Amongst drugs, almost half of the cases are attributed to vaccines, with the measles-mumps-rubella (MMR) vaccine being the most common culprit [5]. With the recent outbreak of the COVID-19 pandemic and emergency use authorization of anti-SARS-CoV2 vaccines, reports of post-vaccine thrombocytopenia have gained attention [6]. Vaccine-induced immune thrombotic thrombocytopenia (VITT) has now been increasingly recognized, predominantly after the administration of adenovirus-vector-based vaccines [7]. It is marked by the formation of widespread thrombi and positive platelet factor 4 antibodies, a lab parameter classically seen in patients with heparin-induced thrombocytopenia. Furthermore, reports of post-vaccine thrombotic thrombocytopenic purpura, defined by low ADAMTS-13 activity and microangiopathic hemolytic anemia, have also emerged [8]. Contrary to these two entities, vaccine-related ITP involves isolated thrombocytopenia and has a relatively favorable prognosis. That said, patients with ITP still carry a higher thromboembolism risk and increased mortality compared to the general population [2]. Therefore, it becomes imperative to understand its epidemiology in relation to the administration of COVID-19 vaccines. With this review, we aim to analyze the clinical characteristics, presenting features, laboratory parameters, therapeutic strategies, and outcomes of patients presenting with ITP after receiving COVID-19 vaccination.

## 2. Materials and Methods

### 2.1. Search Strategy and Selection of Studies

Following the Preferred Reporting Items for Systematic Reviews and Meta-Analyses (PRISMA) guidelines, Medline, Embase, and Ebsco databases were systematically explored from inception until 1 June 2022, with the following keywords: “Purpura, Thrombocytopenic, Idiopathic” (Mesh) AND “COVID-19 Vaccines” (Mesh) OR “2019-nCoV Vaccine mRNA-1273” (Mesh) OR “BNT162 Vaccine” (Mesh) OR “ChAdOx1 nCoV-19” (Mesh) OR “Ad26COVS1” (Mesh). The search was accomplished by two independent authors (PS and NG). Papers hence identified underwent screening at the title and abstract level. The following inclusion criteria were used: 1. case reports and case series investigating the association between the anti-SARS-CoV-2 vaccine and ITP; 2. presence of isolated thrombocytopenia in the absence of thrombosis; 3. ITP is diagnosed after ruling out other causes of thrombocytopenia. Duplicate articles, articles in languages other than English, and studies reporting other causes of thrombocytopenia were excluded. Cohort studies, data from surveillance systems, and review articles were excluded as detailed information on demographics, treatment, and outcome was required for each patient to synthesize the results of this analysis. After reviewing the full text of the eligible articles and overcoming disagreements through discussion, a total of 43 case reports (66 patients) were included in this study (Figure 1).

### 2.2. Data Extraction

We extracted the following data: 1. author name and year of publication; 2. gender of the patient and age at presentation; 3. comorbidities; 4. presenting features; 5. platelet counts at presentation, or nadir if admission counts were not reported; 6. type and dose of the vaccine received; 7. time to symptom onset or presentation post-vaccine, whichever came early; 8. therapies received; 9. outcomes; and 10. whether the second dose, if applicable, was administered or not.

## 3. Results

We found a total of 66 patients with ITP following COVID-19 vaccination, as listed in Table 1. The median age of presentation was 52 years (range: 19–86 years) with a female preponderance (60.6%, *n* = 40). Twenty-four patients had a pre-existing autoimmune disease (17 had pre-existing ITP), one was nine weeks pregnant, and one was receiving immunotherapy (durvalumab) for refractory lung adenocarcinoma. One patient with chronic ITP had a history of flare-up post-Shingrix vaccine. On presentation, two patients had concurrent active Hepatitis C and HIV viral infection, one had autoimmune hemolytic anemia (Evans syndrome), and one patient had weakly positive platelet factor 4 antibodies. Most of the patients (85%) presented with spontaneous mucocutaneous bleeding (gums > nose) or petechiae. Two patients presented with hemoptysis, and none with life-threatening intracranial hemorrhage. The mean time from vaccine administration to symptom onset was 8.4 days, with 73% of patients (*n* = 48) presenting after the first dose and 27% of patients (*n* = 18) after the second dose. More ITP events were triggered by mRNA vaccines (BNT162b2 (*n* = 29) > mRNA-1273 (*n* = 13)) than with adenoviral vaccines (ChAdOx1-S AstraZeneca (*n* = 15) > Ad26.COV2-S Johnson & Johnson (*n* = 9)). On laboratory workup, two patients had positive SS-A antibodies, one had positive GPIb IgG, two had positive lupus anticoagulant, and three had positive GPIIb/IIIa antibodies. A total of 71% of patients (*n* = 47) had thrombocytopenia of ≤10 × 10^3^/μL. Most patients were treated with steroids or IVIG, or both. Escalation of therapy with rituximab and thrombopoietin receptor agonists (TPO-RA) (eltrombopag or romiplostim) was needed in 22 patients (four patients had pre-existing ITP while 18 were newly diagnosed), out of which, in addition, two received vinca alkaloids, two received aminocaproic acid, one received danazol, one received Rho IgG, and one received fostamatinib. Seven patients, all with platelet counts of >30 × 10^3^/μL, were not treated. The overall outcome was promising, with no reported deaths. Ten patients had a relapse, either during hospitalization or post-discharge. One patient had an emergency room visit due to iatrogenic thrombocytosis from treatment (platelet transfusion, IVIG, steroids, TPO-RA, and vincristine during hospitalization). Four patients who developed ITP after the first dose received the second dose of mRNA-based vaccines with no further relapse. A comparison of new cases of ITP versus relapse post-vaccination is illustrated in Table 2.

## 4. Discussion

As of July 2022, over 500 million doses of the COVID-19 vaccine have been delivered across the United States [50]. While there have been around 300,000 reports of adverse outcomes following mRNA vaccination, more than 90% of those were non-serious [51]. Some major adverse events, such as myopericarditis, Guillain-Barre syndrome, and coagulopathy, including ITP, have prompted the need for closer surveillance in the peri-vaccination period [51]. The concept of vaccine-related ITP is not new and has been documented in relation to various other vaccines, such as MMR, influenza, hepatitis B, polio, pneumococcal vaccines, etc. [52]. Most studies reported the occurrence of thrombocytopenia within six weeks of receiving the vaccine and more than 90% of these cases were self-limiting, with only a few progressing to chronic thrombocytopenia [53,54]. With the advent of COVID-19 vaccines, it has become challenging to monitor such cases owing to the emergent need to countermeasure the pandemic, ushering in expedited manufacturing of the vaccines.

Diagnosis of ITP is one of exclusion, involving a thorough history taking to rule out other causes of thrombocytopenia and screening for secondary etiologies of ITP, along with the demonstration of mere thrombocytopenia on peripheral smear without any other hematologic abnormalities [3]. Antiplatelet antibodies are positive in less than two-thirds of patients with ITP, with poor specificity, and are therefore not recommended for diagnosis [55]. In all the cases described above, the authors came to the diagnosis of ITP based on the temporal sequence of events and the absence of any other inciting factors for thrombocytopenia. Despite the question of causality, it is worthwhile to underscore the possible pathophysiological mechanisms by which vaccine-associated ITP might occur (Figure 2). Molecular mimicry, epitope spreading, polyclonal activation, superantigen, and bystander activation are some suggested mechanisms. Analogous to natural infection-causing autoimmunity, both vaccines and their adjuvants carry the structural potential to generate and enhance self-reactivity, respectively [56]. This dysregulated immune response can also lead to the formation of immune complexes, which can additionally perpetuate platelet damage. Furthermore, antisense oligonucleotides, a constituent of the mRNA vaccines, have an inherent ability to cause thrombocytopenia, albeit with a need for a much higher dose than is delivered by a single injection [57]. Possible mechanisms proposed for this effect are platelet consumption through binding of receptors, formation of antibodies on repeated exposure, or an electrostatic platelet-binding effect similar to heparin [57]. Alternatively, a subclinical ITP can manifest as full-blown ITP post-vaccination [6]. Lastly, de novo ITP remains a distinct possibility, particularly in patients developing symptoms some days after vaccination, and is further affirmed by a positive response to traditional ITP-directed therapies, suggesting immune-mediated platelet destruction [6]. The potential mechanism involves vaccine-mediated polyclonal B- and T-cell activation causing both peripheral and bone marrow platelet destruction.

Consistent with natural ITP demographics in patients younger than 65, women were more likely to have vaccine-related ITP, in our review. ITP has long been associated with autoimmune diseases such as systemic lupus erythematosus, thyroid, and inflammatory bowel diseases [58]. A total of 13% of patients in our review had an established prior diagnosis of autoimmune disease, with thyroid disorders being the most common. It is noteworthy that 4.5% (*n* = 3) of patients with no history of autoimmune disorders had evidence of positive autoantibodies on laboratory workup. Given the short follow-up period, the inference of whether these patients had an undiagnosed autoimmune disease, or if antibodies were elevated as a consequence of ITP, is challenging to make. Tarawneh et al. [22] reported the resolution of SSA-antibodies on follow-up, thereby favoring the latter hypothesis. Two-thirds of the post-vaccine ITP cases were seen after mRNA vaccines (64%), and whether this is due to the upregulation of toll-like receptors by mRNA vaccines leading to further immune activation is still unknown [59]. A formal diagnosis of ITP was present in 24% of patients before presentation, highlighting the possible risk of relapse post-vaccination. The mean time to presentation was around eight days, which is consistent with other epidemiological studies [14,60]. A total of 86% of patients presented with thrombocytopenia of less than or equal to 30 × 10^3^/μL, the commonly accepted threshold of ITP patients presenting with major bleeding [61]. It is essential to highlight that there might have been other cases of subclinical ITP that have gone undetected, given the absence of symptoms.

Standard treatment guidelines for ITP favor a short course of steroids as the first-line therapy, with the addition of intravenous immunoglobulin (IVIG) to rapidly increase counts [62]. In our review, most patients showed an improvement in their platelet count and bleeding manifestations with the use of steroids and/or IVIG. Second-line agents were employed in around 33% of patients, which is higher than the use of such agents in a report of over 200,000 ITP patients, wherein only 5% of patients were on such therapeutic modalities [63]. Some patients (*n* = 7) with counts of >30 × 10^3^/μL were not treated per the 2019 recommended guidelines [62]. Except for one reported death from intracranial bleed due to suspected vaccine-related ITP [64], our review showed an excellent short-term prognosis in terms of zero fatalities and safe discharge if hospitalized, for all 66 patients. While data on long-term outcomes of such patients is lacking, prior studies with other vaccine types have shown patients to have a better prognosis than for viral-associated ITP, which is more likely to progress to a chronic state (28% versus 10% following vaccination) [65]. More prospective data are needed to guide patients on the long-term prognosis and chronicity (if any) of anti-SARS-CoV-2 vaccine-related ITP.

Conflicting evidence exists on whether the aforementioned cases were caused by vaccines or were a mere coincidence. As per some reports, the background incidence rates of ITP remained similar in the pre-and post-vaccination periods [6,66]. On the contrary, the administration of the AstraZeneca vaccine in Scotland, Australia, and England showed a higher-than-expected ITP rate [60,67,68]. More prospective data are needed to adjudicate this relationship accurately and identify predictive biomarkers. Whether the next dose should be advised in such patients and whether of the same vaccine type are other potential areas that need exploring. While only four patients in our review received the next dose, reports of safely immunizing chronic ITP patients, in some cases with an alternate vaccine, have been reassuring [67]. Monitoring of platelet counts in the peri-vaccination period, along with sharing the knowledge to present to the hospital emergently upon the first bleeding symptom, are a few of the potential steps that can be taken while administering the next dose.

The main limitations of our study are the absence of confirmatory tests or a standard definition for anti-SARS-CoV-2 vaccine-associated ITP. Even with a decent sample size, comments on the type of association cannot be made as the included studies were case reports. Furthermore, there exists a likelihood of bias in reporting subclinical cases with an emphasis on reporting instances of severe thrombocytopenia. Lastly, there is always a potential of missing germane articles with any review, despite employing a robust search strategy.

## 5. Conclusions

Although vaccine-related ITP is rarely a cause of death, it significantly hampers patients’ quality of life owing to the fatigue and adverse effects of therapeutic interventions, making it a critical pathology to understand in the context of accelerated global vaccination efforts. Our review attempts to make physicians conscious of ruling out this entity while evaluating patients presenting with thrombocytopenia after receiving the anti-SARS-CoV-2 vaccine. In our solicited opinion, the rarity of these events and excellent outcomes for patients should not change views regarding the benefits provided by immunization to combat this global crisis. Instead, it should raise awareness about the need for an in-depth anamnesis before vaccination.

## Figures and Tables

**Figure 1 vaccines-10-01444-f001:**
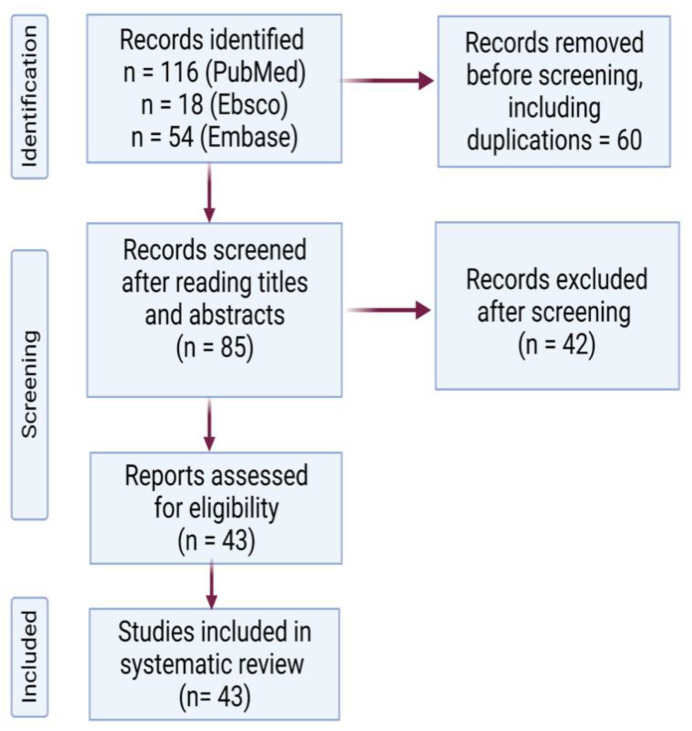
PRISMA flow diagram for selected studies.

**Figure 2 vaccines-10-01444-f002:**
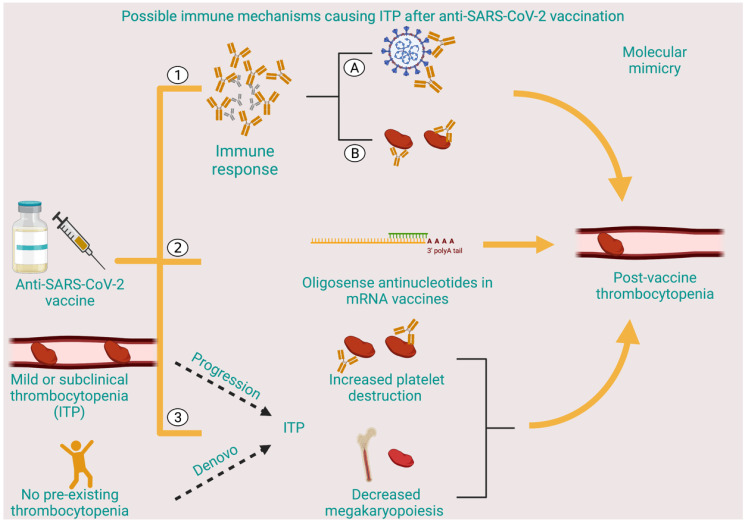
Possible immune mechanisms causing ITP after anti-SARS-CoV-2 vaccination: (1) Immune response causing production of protective antibodies against SARS-CoV-2 (A) with possible molecular mimicry against platelet antigens (B); (2) oligosense antinucleotides in mRNA vaccines causing post-vaccine thrombocytopenia; (3) development of ITP, either de novo or progression from sub-clinical ITP after anti-SARS-CoV-2 vaccine.

**Table 1 vaccines-10-01444-t001:** Characteristics of included studies.

Author, Year	Age	Sex	Presenting Features	Time to Symptom Onset or Presentation after Vaccine (in Days)	Vaccine Dose	Comorbidities	Admission Platelet Counts or Nadir (Whichever Is Reported First)	Treatment Received	Outcomes	Second Dose, If Applicable
Cases due to Ad26.COV2-S (Johnson & Johnson vaccine)										
Shah et al., 2021 [4]	59	F	Abdominal cramps, diarrhea	2	-	Chronic ITP and SLE (prior flare-up 2 years ago after Shingrix vaccine)	64 × 10^3^/μL	Dexamethasone	Discharged	No comment
Banerjee et al., 2021 [9]	63	F	Bleeding gums	14	-	Cervical cancer s/p hysterectomy	2 × 10^3^/μL	Platelets, Prednisone, IVIG, and dexamethasone	Discharged on day 5	-
Cases due to ChAdOx1 nCoV-19 vaccine										
Scanvion et al., 2021 [10]	62	F	Asymptomatic	6	First dose	ITP, overweight, HLD, HTN	60 × 10^3^/μL	None	No hospitalization	No comment
Scanvion et al., 2021 [10]	45	F	Asymptomatic	3	First dose	ITP	53 × 10^3^/μL	None	No hospitalization	No comment
Candelli et al., 2021 [11]	28	M	Petechiae, oral bleed, fatigue, headache, fever	1	First dose	None	4 × 10^3^/μL (Lupus anticoagulant positive)	Four-day Dexamethasone course, once during hospitalization and once 10 days after discharge	Discharged after 4 days	No comment
Paulsen et al., 2021 [12]	72	M	Petechiae, hematomas	11	First dose	Radioiodine-treated autoimmune thyroiditis	<5 × 10^3^/μL	Prednisolone, IVIG	Discharged	No comment
Paulsen et al., 2021 [12]	71	F	Petechiae, headache	11	First dose	Latent hyperthyroidism, breast cancer, stroke	<5 × 10^3^/μL	Prednisolone followed by dexamethasone, IVIG, TPO-RA	Discharged but readmitted 7 days after	No comment
Paulsen et al., 2021 [12]	66	M	Petechiae, hyposphagma	1	First dose	HTN, mild thrombocytopenia	< 5 × 10^3^/μL	Prednisolone	Discharged	No comment
Paulsen et al., 2021 [12]	64	F	Petechiae, epistaxis	15	First dose	HTN, COPD, hepatic steatosis	6 × 10^3^/μL	Prednisolone	Discharged	No comment
Gardellini et al., 2021 [13]	63	M	Hematomas, epistaxis	14	First dose	DM, HTN, HLD	2 × 10^3^/μL	Prednisone	Discharged	Second dose given after 9 weeks
Kim et al., 2021 [14]	66	F	Bruising, gum bleeding	2	First dose	None	4 × 10^3^/μL	Dexamethasone and IVIG	Discharged	No comment
Sivaramakrishnan et al., 2022 [15]	- (middle-aged)	F	Hemoptysis and menorrhagia	11 (had been evaluated 30 days after first dose for similar complaints and was given platelets)	First and second doses	None	10 × 10^3^/μL	Prednisolone	Discharged with relapse after 3 weeks	-
Al-Ahmad et al., 2022 [16]	56	M	Admitted for partial small bowel obstruction	14	First dose	Primary ITP s/p splenectomy	9 × 10^3^/μL	IVIG and TPO-RA (refused steroids)	Discharged with a repeat dose of TPO-RA for worsening counts	No comment
Al-Ahmad et al., 2022 [16]	63	F	Petechiae	10	Second dose	Chronic ITP	35 × 10^3^/μL	None	Not admitted	No comment
Al-Ahmad et al., 2022 [16]	28	F	Petechiae, gum, and nosebleed	10	First dose	Chronic ITP (maintained on romiplostim and prednisolone)	30 × 10^3^/μL	Romiplostim was continued and prednisolone dose was increased	Not admitted	No comment
Al-Ahmad et al., 2022 [16]	54	F	Ecchymoses	13	First dose	None	10 × 10^3^/μL	Prednisolone and IVIG	Discharged after 6 days – readmitted 3 days later with counts of 10 × 10^3^/μL	No comment
Al-Ahmad et al., 2022 [16]	33	F	Ecchymoses	21	First dose	None	3 × 10^3^/μL	IVIG, Prednisolone, and Romiplostim	Discharged after 7 days, was readmitted 26 days later with recurrence	No comment
Wong et al., 2022 [17]	86	M	Gingival bleeding, ecchymosis, and tongue blisters	2	First dose	NA	4 × 10^3^/μL	Dexamethasone, platelets, IVIG and Rituximab	Discharged	NA
Wong et al., 2022 [17]	38	F	Petechiae and purpura	10	First dose	NA	3 × 10^3^/μL	Prednisone and IVIG	Discharged	NA
Razzaq et al., 2021 [18]	26	M	Asymptomatic	2	First dose	Mild thrombocytopenia	64 × 10^3^/μL	Methylprednisolone and IVIG	Discharged	No comment
Uaprasert et al., 2022 [19]	80	M	Bleeding from bitten tongue	19	First dose	None	14 × 10^3^/μL	Dexamethasone, prednisolone, IVIG, TPO-RA	Improved	No comment
Uaprasert et al., 2022 [19]	84	M	Dizziness	9	First dose	Adrenal insufficiency due to adrenal histoplasmosis, cirrhosis, past HBV infection	36 × 10^3^/μL (HCV and HIV positive)	None	Improved	No comment
Uaprasert et al., 2022 [19]	55	F	Purpura and oral bleeding	24	First dose	HLD	41 × 10^3^/μL	None	Improved	No comment
Liao et al., 2021 [20]	79	M	Asymptomatic	8	First dose	Stroke	2 × 10^3^/μL	Hydrocortisone followed by prednisolone	Discharged after 12 days	No comment
Cases due to BNT162b2 vaccine										
Ganzel et al., 2021 [21]	53	M	Epistaxis, purpura, petechiae	14	First dose	Obesity, DM, HTN	1 × 10^3^/μL	Dexamethasone and IVIG	Improved	Second dose not given
Tarawneh et al., 2021 [22]	22	M	Petechiae, gum bleeding	3	First dose	None	2 × 10^3^/μL (mild transaminitis, SSA-antibody which later normalized)	Dexamethasone, platelet transfusion, and IVIG	Discharged after 5 days	No comment
Fueyo-Rodriguez et al., 2021 [23]	41	F	Fever, tachycardia, nausea	1	First dose	HTN, hypothyroidism, pre-DM	65 × 10^3^/μL (elevated IgE and CRP)	Methylprednisolone, dexamethasone, and IVIG	Discharge after 5 days	No comment
Shah et al.,2021 [4]	53	M	Fever, chills, myalgia, petechiae	8	Second dose	Crohn’s disease	2 × 10^3^/μL	Dexamethasone and IVIG	Discharged	No comment
Shah et al.,2021 [4]	67	M	Generalized weakness, melena, petechiae	2	First dose	Seizures, atrial fibrillation, chronic ITP in remission	2 × 10^3^/μL	Platelet, IVIG and dexamethasone	Discharged	Second dose not advised
Jawed et al., 2021 [24]	47	F	Gum bleeding, epistaxis	18	First dose	Chronic ITP, Hypothyroidism secondary to Hashimoto’s thyroiditis	1 × 10^3^/μL	Platelet, IVIG	Discharged	No comment
King et al., 2021 [25]	39	F	Petechiae	1	Second dose	PCOS	1 × 10^3^/μL (elevated ESR)	Platelet, methylprednisolone, and IVIG	Discharged after 3 days	No comment
Gardellini et al., 2021 [13]	27	M	Hematomas, epistaxis	10	First dose	None	1 × 10^3^/μL	IVIG, prednisolon, dexamethasone	Discharged	No comment
Gardellini et al., 2021 [13]	39	F	Petechiae, ecchymosis	6	Second dose	Chronic ITP	1 × 10^3^/μL	IVIG-prednisone, TPO-RA	Not reported	No comment
Qasim et al., 2021 [26]	28	M	Petechiae and epistaxis	2	Second dose	ITP	1 × 10^3^/μL	IVIG and dexamethasone	Discharged on prednisolone taper	Not reporter
Shonai et al., 2021 [27]	69	M	Oral bleeding and hemoptysis	10	Second dose (had asymptomatic thrombocytopenia after first dose)	Well-controlled postoperative intestinal obstruction and hypopharyngeal cancers s/p permanent tracheal fistula surgery	6 × 10^3^/μL (H pylori antibody positive)	Prednisolone	Improved (refused hospitalization)	No comment
Krajewski et a., 2021 [28]	74	M	Hemorrhagic mucosal blisters and purpura	1	First dose	HTN	2 × 10^3^/μL	Platelet and Dexamethasone	No comment	No comment
Al-Ahmad et al., 2022 [16]	19	M	Mouth and nosebleed	4	Second dose	Chronic ITP (maintained on eltrombopag)	4 × 10^3^/μL	Methylprednisolone, prednisolone, and increased dose of eltrombopag	Left against medical advice	No comment
Idogun et al., 2021 [29]	54	F	Petechiae, ecchymosis, mucosal bleeding	7 days after first dose but presented 5 days after second dose (21 days after symptom onset)	First and second doses	HTN, congenital epidermal dysplasia, overactive bladder, mild cognitive impairment, CKD and anxiety	0	Platelet, dexamethasone, IVIG	Discharged but was readmitted after 4 days	-
Hidaka et al., 2022 [30]	53	F	Shortness of breath	14 days after second dose but had transient wheezing and purpura after first dose	First and second doses	Asthma, Vogt-Koyanagi-Harada disease, Hashimoto disease	39 × 10^3^/μL (Also had AIHA, lupus anticoagulant and ANA positive, hypocomplementemia, COVID IgG positive)	Prednisolone, blood transfusion (for Evans syndrome associated with SLE post-COVID vaccination)	Discharged	-
Al-Ahmad et al., 2022 [16]	30	F	Petechiae	7	First dose	Chronic migraine, depression, chronic ITP	40 × 10^3^/μL	None	Not admitted	No comment
Saito et al., 2022 [31]	66	F	Malaise, lymphadenopathy, fever, hematuria, oral bleeding, and purpura	2	First dose	None	<1 × 10^3^/μL (positive antiplatelet glycoprotein IIb/IIIa antibodies, elevated inflammatory markers)	Platelet, IVIG, prednisolone, pulsed methylprednisolone, TPO-RA, danazol and vincristine	Discharged on day 22	No comment
Pasin et al., 2022 [32]	84	M	Petechiae, gum bleeding	5	First dose	Localized bladder cancer, tremors, mild CKD, Atrial fibrillation on apixaban	3 × 10^3^/μL (SARS-CoV-2 negative; positive antiplatelet glycoprotein IIb/IIIa antibodies)	Platelet, IVIG and prednisone	Improved	None
Nakamura et al., 2022 [33]	32	F	Petechiae, oral bleeding	5	Second dose	None	<1 × 10^3^/μL (Platelet associated GPIbα IgG)	Prednisolone	Discharged on day 12	No comment
Al-Ahmad et al., 2022 [16]	37	F	Petechiae	10	Second dose	Primary ITP	25 × 10^3^/μL	Prednisolone	Improved	No comment
Al-Ahmad et al., 2022 [16]	30	M	Fatigue, petechiae, gum bleeding, epistaxis	7	First dose	Primary ITP (on eltrombopag)	11 × 10^3^/μL	Prednisolone, IVIG and eltrombopag	Discharged on day 2	Allowed to take second dose with close follow-up
Al-Ahmad et al., 2022 [16]	56	F	Gum and nose bleeding	7	Second dose	HTN, DM	2 × 10^3^/μL	IVIG, prednisolone and TPO-RA	Discharged after 3 days but readmitted 2 weeks later	No comment
Ogai et al., 2021 [34]	64	F	Oral bleeding and petechiae	2	First dose	Chronic ITP	1 × 10^3^/μL	Prednisolone and IVIG	Improvement	No comment
Ogai et al., 2021 [34]	61	F	Petechiae	17	Second dose	Chronic ITP, Scleroderma and Sjogren syndrome	1 × 10^3^/μL	Platelet, Prednisolone and TPO-RA	Improvement	No comment
Battegay et al., 2021 [35]	77	M	Asymptomatic; petechiae on buccal mucosa	8	First dose	CAD, Atrial fibrillation, HTN (Had mild thrombocytopenia pre-vaccination)	28 × 10^3^/μL	IVIG, prednisone and TPO-RA	Improvement	Second dose under eltrombopag taper
Ghosh et al., 2022 [36]	63	F	Rash and easy bruising	1	Second dose	COPD, HTN, DM	0 (positive for SS-A and scleroderma antibodies)	Dexamethasone, IVIG, TPO-RA, rituximab	Discharged	-
Akiyama et al., 2021 [37]	20	F	Subcutaneous hemorrhage	12	First dose	None	16 × 10^3^/μL	Prednisolone	Improved	No comment
Jasaraj et al., 2021 [38]	67	F	Petechiae, gum bleeding, epistaxis, subconjunctival hemorrhage	2	Second dose (also had symptoms 14 days after the first dose)	HTN, DM, hypothyroidism, depression, b12 deficiency, headaches	3 × 10^3^/μL	Prednisone, IVIG, platelet, ACA, rituximab, TPO-RA	Discharged on day 14 (received rituximab and TPO-RA outpatient)	-
Ghous et al., 2021 [39]	69	F	Bruising and gum bleeding	14	First dose	Cataract, SCC, BCC	5 × 10^3^/μL (AST and LDH were high)	Platelet, IVIG, dexamethasone, TPO-RA, vincristine, prednisone	Discharged with return to ER for iatrogenic thrombocytosis	No comment
Cases due to mRNA-1273 vaccine										
Abuhelwa et al., 2021 [40]	65	F	Epistaxis and rash	1	First dose	None	3 × 10^3^/μL	Platelets, IVIG, dexamethasone, Rho immunoglobulins and TPO-RA	Discharged on day 14	No comment
Prasad et al., 2021 [41]	58	M	Mucosal bleed, petechiae	21	First dose	HTN, DM	3 × 10^3^/μL (PF4 antibody was weakly positive but SRA negative)	Dexamethasone, platelets, IVIG, second relapse treated with platelets, IVIG, TPO-RA and fostamatinib	Discharged in 6 days but presented 5 days later with recurrence, and then again 10 days later	Refused
Ogai et al., 2021 [34]	73	F	Petechiae	11	First dose	HTN, HLD	2 × 10^3^/μL	Prednisolone, IVIG, and TPO-RA	Improvement	No comment
Chanut et al., 2022 [42]	73	F	Epistaxis, intra-buccal hemorrhage, and bruises	7	First dose	IgA monoclonal gammopathy of undetermined significance, HTN, HLD, hypothyroidism, glaucoma	2 × 10^3^/μL	IVIG	Discharged	Rechallenged with BNT162b2 vaccine with no relapse
Helms et al., 2021 [43]	74	M	Epistaxis and diffuse cutaneous purpura	1	First dose	HTN, Gout, HLD and nonischemic cardiomyopathy	10 × 10^3^/μL	Dexamethasone, IVIG, rituximab, TPO-RA	Discharged on day 5; readmitted on day 13	No comment
Chong et al., 2022 [44]	75	F	Hemoptysis	3	First dose	Refractory lung adenocarcinoma on durvalumab	7 × 10^3^/μL (prior hepatitis B infection)	Platelets, prednisolone	Discharged on day 5	Second dose not advised
Malayala et al., 2021 [45]	60	M	Purpura, nausea, vomiting, shortness of breath, leg edema, chest and abdominal pain	1	First dose	Hepatitis C infection, CKD-stage IV, HTN, HFrEF, smoker	84 × 10^3^/μL (With deranged LFTs – heavy hepatitis C viral load and cirrhosis)	None	Left against medical advice on day 3 of hospitalization	No comment
Gardellini et al., 2021 [13]	24	M	Petechiae	21	Second dose	None	2 × 10^3^/μL	IVIG and prednisone	Discharged on steroid taper	No comment
Julian et al., 2021 [46]	72	F	Rash, spontaneous oral bleeding, headache	1	First dose	DM, seasonal contact dermatitis, gout	12 × 10^3^/μL (prior parvovirus infection)	Dexamethasone, IVIG, ACA, rituximab, and platelets	No comment	No comment
Toom et al., 2021 [47]	36	F	Petechiae, easy bruising, bleeding gum, headache	7	First dose	Familial ITP	3 × 10^3^/μL (unlikely due to vaginal estrogen ring)	Dexamethasone and IVIG	Discharged	No comment
Shonai et al., 2021 [27]	34	F	Purpura	21	Second dose	None	11 × 10^3^/μL	Initially not treated, however, on 1 week follow- up, had worsened platelet count for which prednisolone and TPO-RA were given	Improved	-
Hines et al., 2021 [48]	26	F	Petechiae	7	First dose o	Irregular menses on OCPs	19 × 10^3^/μL (transaminitis present)	Prednisone, dexamethasone, and IVIG	Discharged on day 5	No comment
Bennett et al., 2021 [49]	32	F	Petechiae and bruising	11	First dose	None – was currently pregnant at 9 weeks	1 × 10^3^/μL	Prednisone	Discharged on day 3	No comment

ACA: aminocaproic acid; AIHA: autoimmune hemolytic anemia; BCC: basal cell carcinoma; CAD: coronary artery disease; COPD: chronic obstructive pulmonary disease; CKD: chronic kidney disease; DM: diabetes; ER: emergency room; F: female; HFrEF: heart failure with reduced ejection fraction; HLD: hyperlipidemia; HTN: hypertension; HBV: hepatitis B virus; ITP: immune thrombocytopenic purpura; IVIG: intravenous immunoglobulins; M: male; OCP: oral contraceptive pills; PCOS: polycystic ovarian syndrome; SCC: squamous cell carcinoma; SLE: systemic lupus erythematosus; SRA: serotonin release assay; TPO-RA: thrombopoietin receptor agonists, e.g., eltrombopag or romiplostim.

**Table 2 vaccines-10-01444-t002:** Comparison of new cases of ITP versus relapse, post-vaccination.

Variables	Post-Vaccination ITP Flare-Up (*n* = 29)	Post-Vaccination New ITP Diagnosis (*n* = 49)
Median age in years at presentation [42]	43 (19–67)	53 (20–86)
Females (%)	41%	57%
Most common symptom	Mucocutaneous bleed	Mucocutaneous bleed
Number of asymptomatic individuals	3	3
Median days to presentation	7.5	12.5
Number of cases/vaccine	1—Ad26.COV2-S (Johnson & Johnson vaccine) 5—ChAdO × 1 nCoV-19 vaccine 10—BNT162b2 vaccine 1—mRNA-1273 vaccine	1—Ad26.COV2-S (Johnson & Johnson) vaccine 18—ChAdO × 1 nCoV-19 vaccine 18—BNT162b2 vaccine 12—mRNA-1273 vaccine
Median platelet counts	32.5 × 10^3^/μL	42 × 10^3^/μL
% of patients needing escalation of treatment with second-line agents	10%	32.6%

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
