# Peer review of "A Systematic Review of Reported Cases of Immune Thrombocytopenia after COVID-19 Vaccination"

_vaccines, 2022, doi:10.3390/vaccines10091444_

Round 1

Reviewer 1 Report

Prachi Saluja et al. aimed to perform a systematic review of the reported cases of ITP after COVID-19 vaccination. Undoubtedly, a topic of interest.

I have some questions/comments as follow:

 1.     The way of selecting the literature, neglecting major publications that contributed to the topic, requires better explanation.

2.     Line 88, age: since the authors report range they should show median instead of mean.

3.    Line 89, pre-existing diseases. They way to group the diseases should be re-evaluated. ITP is an autoimmune disease; also the mentioned autoimmune hemolytic anemia. All these diseases should be reported as autoimmune diseases (afterwards subgroups can be mentioned).

4.     Bleedings: percentage of life-threatening bleedings should be reported.

5.     Line 96, “ the mean time from vaccine administration to symptom”: I would suggest to report median (range) time.  

6.  Line 97, 48 patients presenting after the first dose and 18 patients after … (add percentage).

7.     Report the number (and %) of patient with Platelet count below 10.

8.   Line 103, therapy escalation: provide in which type of patient it was necesarry, previous diagnosed ITP or in the novo.

9.  A table comparing cases of pre-existing ITP  to the novo cases would be illustrative.

10.  Line 170, “excellent clinical response”: what is an excellent clinical response in ITP? Specify please.

11. Line 176 “an excellent short-term prognosis for ….”. What is an excellent short-term prognosis? Being discharged from the hospital? Specify please.

12.  The discussion should be more careful and it should mention the difficulties of how to discuss the prognosis with these patients, since some of them will evolve into chronicity.

13.  Perhaps a couple of recommendations on how to monitor these patients in case they get a COVID vaccine again would be helpful for the readers.

Author Response

Dear Editors of Vaccines journal

We are re-submitting our manuscript entitled " A Systematic Review of Reported Cases of Immune Thrombocytopenia after COVID-19 Vaccination," Manuscript ID: vaccines- 1834633, to be considered for publication in ‘Vaccines journal.' We thank the reviewers for their insightful and instructive comments. We would like to address each reviewer's comments point by point:

Reviewer #1 Prachi Saluja et al. aimed to perform a systematic review of the reported cases of ITP after COVID-19 vaccination. Undoubtedly, a topic of interest.

I have some questions/comments as follow:

Point 1.     The way of selecting the literature, neglecting major publications that contributed to the topic, requires better explanation.

We express our sincere gratitude to Reviewer #1, who took the time to review our article and provide valuable feedback. We have acknowledged this in 2.1. Search Strategy and Selection of Studies section of the review: Cohort studies, data from surveillance systems, and review articles were excluded as detailed information on demographics, treatment, and outcome was required for each patient to synthesize the results of this analysis. (Lines 74-76)

Point 2.     Line 88, age: since the authors report range they should show median instead of mean. 

We want to thank the reviewer for the comment. As pointed out, we have revised it to median age: The median age of presentation was 52 years (range: 19-86 years) with a female preponderance (60.6%, n=40). (Line 82)

  1. Line 89, pre-existing diseases. They way to group the diseases should be re-evaluated. ITP is an autoimmune disease; also the mentioned autoimmune hemolytic anemia. All these diseases should be reported as autoimmune diseases (afterwards subgroups can be mentioned). 

We convey our gratitude to Reviewer #1 for another thought-provoking comment. We have hence modified it to the following: Twenty-four patients had a pre-existing autoimmune disease (17 had pre-existing ITP), one was nine weeks pregnant, and one was receiving immunotherapy (durvalumab) for refractory lung adenocarcinoma. (Lines 93-95)

  1. Bleedings: percentage of life-threatening bleedings should be reported. 

We thank the reviewer for this comment. We have included the following line: Two patients presented with hemoptysis, and none with life-threatening intracranial hemorrhage. (Lines 100-101)

  1. Line 96, “ the mean time from vaccine administration to symptom”: I would suggest to report median (range) time.  

We express our gratitude to the reviewer for the comment. While comparing with other published literature that has reported mean time to presentation in the discussion section, we purposely reported the mean for easy comparison. However, we have reported the median time in table 2 (comparing pre-existing cases to denovo cases).

  1. Line 97, 48 patients presenting after the first dose and 18 patients after … (add percentage). 

We acknowledged this as follows: The mean time from vaccine administration to symptom onset was 8.4 days, with 73% of patients (n = 48) presenting after the first dose and 27% of patients (n = 18) after the second dose. (Lines 101-103)

  1. Report the number (and %) of patient with Platelet count below 10.

We modified the article to report the above: 71% of patients (n = 47) had thrombocytopenia of £10 ×103 /μl. (Line 108)

  1.  Line 103, therapy escalation: provide in which type of patient it was necesarry, previous diagnosed ITP or in the novo. 

We thank the reviewer for this comment. We have made necessary changes to the review to reflect this point: The majority of the patients were treated with steroids and/or IVIG. Escalation of therapy with rituximab, aminocaproic acid, thrombopoietin receptor agonists (TPO-RA) (eltrombopag or romiplostim) and/or danazol was needed in twenty-two patients (4 patients had pre-existing ITP while 18 were newly diagnosed), out of which, in addition, two received vinca alkaloids, one received Rho IgG, and one received fostamatinib. (Lines 109-124)

  1. A table comparing cases of pre-existing ITP  to the novo cases would be illustrative. 

We thank Reviewer #1 for taking the time to appraise our article and provide us with interesting questions to ponder. We have now included table 2 in our article after table 1 to illustrate the same.

  1. Line 170, “excellent clinical response”: what is an excellent clinical response in ITP? Specify please.

As correctly pointed out by reviewer #1, we have specified it as follows: In our review, most patients showed an improvement in their platelet count and bleeding manifestations with steroids and/or IVIG. (Lines 274-275)

  1. Line 176 “an excellent short-term prognosis for ….”. What is an excellent short-term prognosis? Being discharged from the hospital? Specify please.

We convey our gratitude to reviewer #1 for another insightful comment and have hence specified it as follows: Except for one reported death from intracranial bleed due to suspected vaccine-related ITP [64], our review showed an excellent short-term prognosis in terms of zero fatalities and safe discharge if hospitalized, for all 66 patients. (Lines 279-282)

  1. The discussion should be more careful and it should mention the difficulties of how to discuss the prognosis with these patients, since some of them will evolve into chronicity.

We acknowledged this point in our review as follows: More prospective data are needed to guide patients on the long-term prognosis and chronicity (if any) of anti-SARS-CoV-2 vaccine-related ITP. (Lines 289-290)

  1. Perhaps a couple of recommendations on how to monitor these patients in case they get a COVID vaccine again would be helpful for the readers.

We have made necessary changes to the review to reflect this point: Monitoring of platelet counts in the peri-vaccination period, along with sharing the knowledge to present to the hospital emergently upon the first bleeding symptom, are few of the potential steps that can be taken while administering the next dose. (Lines 300-303)

Reviewer 2 Report

In this invited article, the authors present a systematic review of reported cases of immune thrombocytopenia (ITP) after COVID-19 vaccination. Following a search according to the PRISMA guidelines, 66 cases were identified. Based on the analysis of the clinical characteristics, presenting features, laboratory parameters, therapeutic strategies and outcomes of the patients, the authors observed that most ITP events were triggered by mRNA vaccines within about 1 week after vaccination and were treated successfully with administration of steroids and/or intravenous immunoglobulins. A model of possible immune mechanisms causing ITP after COVID-19 vaccination is also discussed.

This article presents a well-written, concise and informative critical review of the literature. It is likely to increase the awareness among clinicians of COVID-19 vaccine-related ITP and the investigation of its possible immune mechanisms.

Line 48-52: has the association of vaccine-induced ITP with the presence of platelet factor 4 (PF4) antibodies and/or low ADAMS13 activity and microangiopathic hemolytic anemia been observed in patients with ITP after COVID-19 vaccination?  If so, indicate those cases in the text and the table.

Line 102-106: please describe the treatment regimens in more detail (dose, route, duration, etc.).

Line 110: after which treatment did iatrogenic thrombocytopenia develop?

Table 1: to facilitate comparing the cases, please organize them by vaccine type and dose. Also, avoid breaking the description of cases over two pages.  List all abbreviations and acronyms in alphabetical order.

Line 142-144: please elaborate as to why mRNA vaccines have an inherent ability to cause thrombocytopenia.

Line 145-148: please elaborate as to possible mechanism(s) of denovo ITP after COVID-19 vaccination.

Author Response

Reviewer #2 In this invited article, the authors present a systematic review of reported cases of immune thrombocytopenia (ITP) after COVID-19 vaccination. Following a search according to the PRISMA guidelines, 66 cases were identified. Based on the analysis of the clinical characteristics, presenting features, laboratory parameters, therapeutic strategies and outcomes of the patients, the authors observed that most ITP events were triggered by mRNA vaccines within about 1 week after vaccination and were treated successfully with administration of steroids and/or intravenous immunoglobulins. A model of possible immune mechanisms causing ITP after COVID-19 vaccination is also discussed.

This article presents a well-written, concise and informative critical review of the literature. It is likely to increase the awareness among clinicians of COVID-19 vaccine-related ITP and the investigation of its possible immune mechanisms.

Line 48-52: has the association of vaccine-induced ITP with the presence of platelet factor 4 (PF4) antibodies and/or low ADAMS13 activity and microangiopathic hemolytic anemia been observed in patients with ITP after COVID-19 vaccination?  If so, indicate those cases in the text and the table.

We express our sincere gratitude to Reviewer #2 for taking the time to appraise our article and provide us with valuable suggestions. While positive PF4 antibodies with thrombocytopenia point toward a diagnosis of vaccine-induced thrombotic thrombocytopenia (VITT) and low ADAMTS13 activity with hemolytic anemia and thrombocytopenia point toward thrombotic thrombocytopenic purpura (TTP), our aim with this review was to elaborate solely on isolated thrombocytopenia post-vaccination. We did, however, include one case that reported weakly positive PF-4 antibodies and one case that developed Evans syndrome in the context of pre-existing ITP. Both cases have been mentioned in text and table 1. On presentation, two patients had concurrent active Hepatitis C and HIV viral infection, one had autoimmune hemolytic anemia (Evans syndrome), and one patient had weakly positive platelet factor-4 antibodies. (Lines 96-99)

Line 102-106: please describe the treatment regimens in more detail (dose, route, duration, etc.).

We thank the reviewer for the thoughtful comment and excellent suggestion. While it would be interesting to comment on each specific treatment modality in another review article, we feel that our aim with this review article was to make physicians conscious of such an entity in the peri-vaccination period and guide physicians on the outcomes of such patients. Most treatment regimen doses stemmed from the existing ITP guidelines, and it is beyond the scope of this review to mention the dose, route, and duration of treatments for 66 patients, the majority of who received >1 treatment.

Line 110: after which treatment did iatrogenic thrombocytopenia develop?

We thank the reviewer for this question. We understand the question meant iatrogenic thrombocytosis, and to reflect the same, we have included the following in the article: One patient had an emergency room visit due to iatrogenic thrombocytosis from treatment (platelet transfusion, IVIG, steroids, TPO-RA, and vincristine during hospitalization). (Lines 127-129)

Table 1: to facilitate comparing the cases, please organize them by vaccine type and dose. Also, avoid breaking the description of cases over two pages.  List all abbreviations and acronyms in alphabetical order.

We thank the reviewer for another excellent suggestions. We have modified table 1 to reflect all these changes.

Line 142-144: please elaborate as to why mRNA vaccines have an inherent ability to cause thrombocytopenia.

We thank the reviewer for the insightful question. We have included the following in our review article: Possible mechanisms proposed for this effect are platelet consumption through binding of receptors, formation of antibodies on repeated exposure, or an electrostatic platelet-binding effect similar to heparin [57]. (Lines 248-251)

Line 145-148: please elaborate as to possible mechanism(s) of denovo ITP after COVID-19 vaccination.

We express our gratitude to the reviewer for the comment. We added the following line: Potential mechanism involves vaccine-mediated polyclonal B-and T-cell activation causing both peripheral and bone marrow platelet destruction. (Lines 255-256)

Round 2

Reviewer 1 Report

The authors have answered my questions, and improved the quality of the manuscript.

Author Response

We thank the reviewer for their comments and for improving the quality of our review paper.